# Exploring Consumer Behavior and Preferences towards Edible Mushrooms in Slovakia

**DOI:** 10.3390/foods12030657

**Published:** 2023-02-03

**Authors:** Kristína Predanócyová, Július Árvay, Marek Šnirc

**Affiliations:** 1AgroBioTech Research Centre, Slovak University of Agriculture in Nitra, 949 76 Nitra, Slovakia; 2Institute of Food Sciences, Faculty of Biotechnology and Food Sciences, Slovak University of Agriculture in Nitra, 949 76 Nitra, Slovakia

**Keywords:** edible mushrooms, consumer, consumption, preference, behavior

## Abstract

Edible mushrooms represent a food with high nutritional properties, and their consumption has a positive effect on the health of consumers. The aim of the paper is to identify the behavior and preferences of consumers in the consumption of edible mushrooms in Slovakia. The aim of the paper was achieved by conducting a consumer survey using the snowball sampling method on a sample of 1166 respondents in the Slovak Republic, of which 1032 respondents were consumers of edible mushrooms. Using statistical methods (the Chi-square test of independence, Kruskal–Wallis H test, and Friedman test, as well as categorical principal component analysis), differences in consumer behavior were examined in three identified segments created based on the amount of consumption of edible mushrooms. The results of the consumer study showed the existence of statistically significant differences between the defined segments in terms of frequency of consumption, evaluation of the preference of edible mushrooms in various meals, evaluation of important reasons for consumption, and determination of preference for individual species of mushrooms, as well as determination of preference for the place of consumption and the option of obtaining mushrooms for consumption. Moreover, four latent components determining the purchase of mushrooms applicable in all segments were defined. Supporting the consumption of edible mushrooms among Slovak consumers is possible by increasing consumer awareness through recommendations and published articles. The research paper provides a new insight into the behavior and preferences of consumers in mushroom consumption, divided into three segments, which can fill the scientific research gap. The results provide valuable information for scientific purposes, as well as for food companies and policy makers.

## 1. Introduction

Edible mushrooms have an important position in the diet of consumers all over the world [1]. The reason is that mushrooms have a high nutritional value and are an excellent source of proteins, glucans, fiber, vitamins, and minerals [2,3]. Mushroom consumption has a positive effect on the health of consumers due to the fact that mushrooms offer prevention of inflammation and detoxification of cancer-triggering compounds [4]. Mushrooms also have a positive effect on strengthening the immune system and reduce the risk of obesity, cancer, and cardiovascular disease [5]. Mushrooms are considered a healthy food that is suitable for consumption by all age generations and are consumed by children and adults all over the world [6,7,8]. In the context of the mentioned health benefits and the composition of mushrooms, the interest in mushrooms may be higher, as they are a suitable alternative to meat, and in the future, they may become a suitable food for people who are looking for a healthy and ecological diet [9]. Mushrooms are also a suitable food for vegetarians and vegans, as they are considered a vegetable and are now consumed worldwide for their nutritional, medicinal, and economic benefits [5,10,11].

Mushrooms have been eaten for many years and have been considered a valuable source of nutrition and medicine. Currently, the consumption of mushrooms is growing in popularity again, and the demand for mushroom-based foods is constantly growing [4]. In recent years, the demand for mushrooms has increased mainly due to their medicinal effects, a healthy lifestyle, and the preference for veganism and vegetarianism [9]. In the context of the above, it is possible to point out that the worldwide consumption of mushrooms is growing rapidly [12], and in recent decades, there has been an increase in the consumption of mushrooms in the United States [13], India [14], and 13 European countries [15]. The consumption of edible cultivated mushrooms, but especially edible wild mushrooms, also has an opposite negative side. In the process of fructification, there is an extreme transfer of risky and potentially risky elements (but also organic contaminants) from the mycelium to the fructifying organs. The mentioned may, depending on the species (genus) of the mushroom, the locality of collection, and other factors, represent a risk of increased passage of hazardous substances into the consumer’s body. Mushrooms collected in locations that are negatively affected by intensive industrial activity [16,17,18,19] and intensive transport [20] are extremely risky.

The consumption of mushrooms by consumers has been the subject of research in several studies, and the results have shown that the majority of consumers consume edible mushrooms [8,21,22]. Adedokun and Okomadu [22] further found that more than half of consumers consume mushrooms only occasionally. The frequency of mushroom consumption can also be estimated based on the frequency of their purchase. Shirur and Shivalingegowda [23] identified that most consumers buy mushrooms only occasionally, once a month or every two months, and only 20% of consumers buy mushrooms weekly or biweekly. Wen et al. [24] identified that the frequency of buying mushrooms, as well as their volume, are relatively low among consumers.

In consumer behavior, it is necessary to point out the preferred species of edible mushrooms. The results of the study conducted by Boin and Nunes [21] showed that canned mushrooms are more preferred for consumption than fresh, dried, or frozen. A study conducted by Owosu and Dekagbey [5] also reached similar results and emphasizes the preference for consuming processed mushrooms. On the other hand, however, Mayett et al. [25] emphasize the high predominance of fresh mushrooms, followed by canned mushrooms and wild mushrooms, in consumer preferences. Haimid et al. [26] add that consumers most often buy fresh mushrooms compared to other processed mushrooms. Thilakarathne and Sivashankar [7] found that the majority of consumers consumed wild mushrooms. Shah [14] identified that the most preferred species of mushrooms are white button mushroom and oyster mushroom, and the least preferred for consumption are paddy straw mushroom, milky mushroom, and shiitake mushroom. The highest preference for button mushrooms was also confirmed by other conducted research, the results of which showed that consumers most often consume white button mushroom compared to other species of mushrooms [5,7,27]. Linde et al. [28] found that the white button mushroom is the most widespread among consumers, followed by oyster mushroom, almond mushroom, and shiitake mushroom. Mayett et al. [25] identified *Agaricus* spp. mushrooms as the most widespread mushroom species for consumption, while *Pleurotus* spp. and *Lentinula edodes* can be classified as the least preferred mushroom species.

In general, it can be stated that mushrooms are consumed mainly because of their taste, nutritional advantages, and health benefits. Christy et al. [29] identified taste and nutritional and therapeutic effects as key motives for eating edible mushrooms. Bringye et al. [9] emphasized that medicinal and functional properties are important for the consumption of mushrooms and pointed out that mushrooms have a positive effect on the immune system, anti-tumor effects, beneficial effects on health, and antiviral and antibacterial effects, and also they are a source of vitamin D, a suitable component of meals. Furthermore, in connection with consumption, they pointed to consumption for enjoyment and emphasized the taste and aroma of mushrooms, the variety of methods of preparation, or the possibility of using them as spices. Mushrooms can also be consumed because they represent a supplementary food source and are an alternative to meat or a suitable ingredient in meals due to their chemical and nutritional composition. Oguntoye et al. [8], in connection with the reasons for consumption, states that most consumers consume mushrooms because of their nutritional value and organoleptic characteristics.

In the context of buying edible mushrooms and their subsequent consumption, it is important to point out the possible determinants. Several studies dealt with the key factors influencing the quantity of mushroom purchases, as well as the amount of consumption. The results of the study conducted by Shah [14] showed that taste is the most important factor in the consumer’s decision to buy mushrooms, followed by brand, which is associated with trust, and the third key factor is packaging. On the other hand, the least important aspect for buying and consuming mushrooms is their price. Shirur et al. [30] found that consumers are most influenced by the color, size, and shape of the mushroom and least influenced by price. If color, size, and shape represent the quality of mushrooms, then quality is perceived as more important than price when buying mushrooms. Mushroom purchase patterns are also determined by smells, pesticide residues, product attributes, packaging, health aspect, safety, place of purchase, branding, and whether there are kids in family or not [24]. Other factors that may influence the purchase and consumption of mushrooms are household size, income, the price of complementary product, the price of mushrooms, and the nutritional benefits of mushrooms [10]. Boin and Nunes [21] identified in their study that educational level, gender, and age also have a significant effect on mushroom consumption. Another important determinant is consumer awareness regarding the consumption of mushrooms and their health benefits [8].

On the other hand, the results of other studies showed that approximately 50% of consumers do not consume edible mushrooms. In the case of no or only occasional consumption of edible mushrooms, several reasons have been identified. The key reasons include the unavailability of mushrooms on the market, the higher price of mushrooms, and lack of knowledge of mushroom recipes, as well as insufficient consumer awareness of nutritional benefits, the freshness of purchased mushrooms, and the unpopularity of mushroom consumption among family members [14,30]. Adedokun and Okomadu [22] state that the main barriers to mushroom consumption are seasonal availability, the shelf life of mushrooms, and financial limitations. In order to make the consumption of mushrooms more attractive to consumers, it is possible to add mushrooms to various meals, as they are characterized by a distinct taste and nutritional values [31]. Mushrooms are most often consumed as part of mushroom-based products, such as fried mushrooms, burgers, pastries, nuggets, popcorn, pickles, biscuits, sauces, soup powder, and candy [29,32], or it also can be used as an ingredient in curry meals or satay [33].

The consumption of mushrooms is also determined by the level of consumer awareness regarding the consumption of mushrooms and their nutritional and medicinal properties. More than 60% of respondents are aware of mushrooms’ low energy content and properties regarding anti-aging, cancer prevention, heart support, strengthening immunity, and nutritive value [8]. However, several studies show that many consumers do not have sufficient consumer awareness, especially regarding nutritional values [8,14,25,30]. However, Shah [14] also states that it is possible to increase consumer awareness of mushroom consumption regarding nutritional aspects through appropriate communication using various marketing channels. It is also possible to increase the purchased and consumed amounts of edible mushrooms through the implementation of various programs focused on the health benefits of mushrooms [34].

For the future of the edible mushroom market, it is essential to know the consumer and purchase behavior and the determinants influencing the consumption of edible mushrooms. On this background, the aim of the paper is to explore consumer behavior in the market of edible mushrooms and to identify consumer preferences for the consumption of edible mushrooms in the Slovak Republic. Currently, consumer behavior is changing in the food market, including edible mushrooms, with an emphasis on lifestyle, healthy eating, organic eating, and alternative diets. The Slovak Republic, as well as other Central European countries with a predominantly Slavic population, belong to the group of countries referred to as “mycophilic” countries [35], but a consumer study focused on the consumption of edible mushrooms is missing in Slovakia and, thus, can fill a scientific research gap. The results of the study are based on a consumer survey conducted in the Slovak Republic on a sample of 1166 respondents, of which 1032 were consumers of edible mushrooms. The study examines the consumer behavior of edible mushroom consumers, divided into segments in terms of the amount of consumption, their frequency of consumption, preferences for individual species of mushrooms, reasons for consumption, place of consumption, options for obtaining mushrooms for consumption, and consumer behavior in the process of mushroom collection and purchase, with an emphasis on the factors determining the purchase and subsequent consumption of mushrooms. The consumer study provides information applicable in the practice of food companies in the placement and sale of mushrooms, as well as in the application of marketing tools necessary for communication with consumers. The presented study also contributes to the literature on the behavior of Slovak consumers in the process of buying and consuming edible mushrooms, as well as on determinants of consumer acceptance and use of edible mushrooms. The study is also beneficial for consumers and provides relevant information for better orientation in the market of edible mushrooms, with the aim of increasing their consumption.

Based on the above reasoning and the aim of this paper, the following research questions were formulated:

RQ1: What are the differences in the behavior and preferences of edible mushroom consumers in individual segments?

RQ2: What are the key factors determining the purchase and subsequent consumption of edible mushrooms?

## 2. Materials and Methods

The study is based on a consumer survey, the aim of which was to explore the behavior of edible mushroom consumers, with an emphasis on the amount of mushrooms consumed. The purpose of the survey was to identify the edible mushroom consumption patterns and preferences among consumers.

The consumer survey was carried out by an online questionnaire survey on a sample of 1166 respondents in the Slovak Republic in 2022. The snowball method was used for data collection, which is considered a standard method of data collection in qualitative research [36,37]. Respondents involved in the questionnaire survey were divided according to selected socio-demographic characteristics, namely, gender, age, education, place of residence, economic status, and number of members in the household (Table 1). In order to fulfill the purpose of the paper, only respondents who are consumers of edible mushrooms were selected. These respondents were divided according to selected socio-demographic characteristics (Table 1).

Edible mushroom consumers involved in the consumer survey were divided into segments based on the consumed amount of edible mushrooms per year. This amount was determined by the consumers themselves. Accordingly, three segments of mushroom consumers were identified: occasional consumers, regular consumers, and heavy consumers. Subsequently, the consumer patterns and preferences of consumers in individual segments were explored, and the differences between individual segments were identified, as well.

Consumers rated the frequency of consumption and chose one of the following options: several times a week, 1 time a week, 1–2 times a month, and occasionally. Furthermore, consumers determined their preference for eating edible mushrooms in individual meals, namely, appetizer, soup, main meal, side dish, salad, and salty snacks. They determined their preference for individual meals containing mushrooms on a 5-point Likert scale, with 1 representing no preference and 5 a very strong preference. In the next part of the survey, consumers determined the key reason for consumption, where they had a choice of five possible reasons, namely, taste, habit/custom, recommendation, part of a rational diet, and lifestyle. An important part was the identification of the preference of individual species of mushrooms, where consumers evaluated 19 selected species of edible mushrooms on a 5-point scale, with 1 representing no preference and 5 a strong preference. Then, the most preferred place for eating edible mushrooms was also examined, and consumers determined one of the four selected places, namely, at home, canteens at school/work, restaurants, and at family members’ homes. A significant part of the survey was the determination of the option of obtaining edible mushrooms for consumption. Consumers chose which of the following options of obtaining mushrooms for consumption they use: cultivation of edible mushrooms, collection of edible mushrooms, buying edible mushrooms, and getting edible mushrooms as a gift. The survey was more focused on identifying the behavior of consumers when collecting and buying mushrooms. In the part of the survey focused on mushroom collection, questions related to the frequency of collection, the choice of collection places, preferred species, and the amount of collected mushrooms included. The purchasing behavior of consumers included questions related to the frequency of purchase, preference for species of mushrooms, acceptability of the price per kg of edible fresh mushrooms, and factors determining the purchase and subsequent consumption of edible mushrooms. Consumers rated a total of 18 factors on a 5-point Likert scale, with 1 representing an insignificant factor and 5 representing a very important factor. The survey was completed by evaluating the influence of information sources to increase and maintain the consumption of edible mushrooms. Consumers evaluated 15 different information sources on a 5-point Likert scale, where 1 represented an insignificant influence of the source and 5 represented a very significant influence of the source.

Differences between segments in the importance of characteristics related to consumer behavior regarding the consumption of edible mushrooms were analyzed using the Chi-square test of independence and the Kruskal–Wallis test. The differences between the evaluated aspects related to consumer preferences in individual segments were analyzed using the Friedman test and the Nemenyi post-hoc test. The factors determining the purchase of edible mushrooms were divided into latent components based on a categorical principal component analysis (CATPCA), which can be applied to a Likert scale or other measures in a questionnaire survey [38]. Analyses were carried out in statistical software XLSTAT 2022.4.1 (Addinsoft, NY, USA) and IBM SPSS Statistics Grad Pack 28.0 (IBM Corp., Armonk, NY, USA). For statistical testing, the significance level was set to 0.05.

## 3. Results

The results of the study showed that 88.5% of consumers involved in the consumer survey consume edible mushrooms, of which more than 40% of respondents indicated only occasional consumption. In terms of quantity of consumption, 26.5% of mushroom consumers consume up to 1.0 kg of edible mushrooms per year, and 49.0% of mushroom consumers consume 1.1 to 5.0 kg of edible mushrooms annually. Furthermore, we identified that 18.2% of mushroom consumers consume from 5.1 to 10.0 kg of mushrooms per year, and 7.3% of mushroom consumers consume more than 10.0 kg of edible mushrooms per year. In terms of the determined annual amount of mushrooms consumed by mushroom consumers, we divided these consumers into three segments. The first segment represents occasional consumers (annual consumption less than 1.0 kg), the second segment represents regular consumers (annual consumption between 1.1 kg and 5.0 kg), and the third segment represents heavy consumers (annual consumption higher than 5.1 kg). From the point of view of the socio-demographic structure of the individual segments, it is possible to state the different composition of the segments (Table 2). The first segment “occasional consumers” is mainly represented by women (63.7%), consumers under the age of 25 years (37.4%), and consumers of mushrooms from the countryside (53.1%). The second segment “regular consumers” includes mostly women (53.3%), consumers of all age categories, but especially 25–39 years old (29.2%), and consumers from cities (55.3%). In the last segment, “heavy consumers”, women (48.4%) and men (51.6%) are approximately equally represented, and compared to the other segments, there is the highest representation of consumers in the age category 40–54 (30.5%) and consumers of the older generation over 54 (20.7%). The segment “heavy consumers” consists mainly of consumers from cities (62.5%). All segments are characterized by the representation of consumers from households with 3–4 members (57.1%) and consumers with higher education who are still students or employed.

Although mushrooms are not among the daily consumed foods, we examined the frequency of their consumption and, based on the results of the applied Chi-square test, we identified statistically significant differences between the individual segments (*p* < 0.0001). The frequency of consumption of edible mushrooms by consumers from individual segments is specified in more detail in Table 3, and it can be stated that in the “occasional consumer” segment with consumption up to 1 kg, occasional consumption (75.5%) and consumption once or twice a month (23.4%) were indicated. In the “regular consumers” segment with an annual consumption level of 1.1 to 5 kg of mushrooms, consumers consume mushrooms at least once or twice a month (48.7%) or occasionally (37.0%). The most frequent consumption of edible mushrooms was identified in the “heavy consumers” segment with an annual consumption of over 5 kg, and based on the results, we conclude that 48% consume mushrooms at least once or twice a month, 30.9% of the consumers consume mushrooms once a week, and almost 10% of the consumers consume mushrooms several times a week. This can also be justified by the fact that this segment includes 9.8% of consumers who do not consume meat or products of animal origin and, thus, replace these foods with other alternatives. 

The frequency of consumption of edible mushrooms, as well as their total amount, can also be influenced by the incorporation of mushrooms into the consumed meals. Consumers rated their preference for mushroom consumption in individual meals on a 5-point Likert scale, with 1 representing no preference and 5 a strong preference. Based on the results of the applied Kruskal–Wallis test, it is possible to state statistically significant differences in the consumption of mushrooms in all meals between individual segments (*p* < 0.0001). Mushrooms as a main meal are most consumed by consumers from the segment “heavy consumers” and the segment “regular consumers”. Consumers from the “occasional consumers” segment mostly consume mushrooms as part of soup. On the other hand, we also identified the least preferred mushroom dishes, and it could be concluded that salty snacks are the least acceptable and preferred for consumers from all segments. In the context of evaluating the preference for eating meals containing mushrooms, we also examined the differences in the evaluation of the preference for selected meals within individual segments. The results of the applied Friedman test showed statistically significant differences in the consumption of mushroom dishes in each segment (*p* < 0.001). Based on the application of the Nemenyi test, we identified differences in the evaluation of preferences for the consumption of these meals, and we can conclude that consumers from the “occasional consumers” segment most often consume mushrooms as part of soup and main meals, followed by flavoring of meals and side dishes, and they are least consumed in salads, appetizers, and salty snacks. Consumers from the “regular consumers” segment prefer the consumption of mushrooms as part of the main meal or soup, the consumption of mushrooms as part of the flavoring of meals and side dishes is less preferred, and the least popular are mushrooms in appetizers, salads, and other snacks. Consumers from the “heavy consumers” segment prefer the consumption of mushrooms in main meals and soups, followed by flavoring of meals and side dishes, and they least prefer mushrooms in appetizers, salty snacks, and salads. The Nemenyi test shows the specification of differences in preference for mushroom consumption in selected meals in individual segments. (Table 4, Table 5 and Table 6).

The amount of consumption can also be influenced by the reason why consumers consume edible mushrooms. Therefore, the key reasons for the consumption of edible mushrooms and the differences in assessment between individual segments were examined. Based on the results of the Chi-square test, statistically significant differences between the segments were identified (*p* < 0.0001). The results show that in the “occasional consumers” segment, almost 70% of consumers consider taste to be the key reason for consumption, followed by habit (19.8%) and rationality in eating (9.2%). In the “regular consumers” segment, taste was the main motivation for mushroom consumption (64.6%), but many consumers also identified habit (16.7%) and rationality in eating (14.5%) as important reasons for consumption. Consumers from the “heavy consumers” segment considered taste (68.4%) and habit/habit (15.2%) as the main motive for consumption. In the mentioned segment, rational eating (8.6%) and lifestyle (7.8%) were also identified as important reasons for consumption. The reasons for consumption identified by consumers from the individual segments are specified in more detail in Table 7, where differences in evaluation are also visible.

Based on the achieved results related to the reason for consumption of edible mushrooms and taste as a key factor, it can be assumed that the species of mushroom can be considered as a determining aspect of consumption. For this reason, the preference of selected species of mushrooms was explored, and based on the results of the Kruskal–Wallis test, statistically significant differences were identified in all examined species of mushrooms (*p* < 0.05). In the “occasional consumers” segment, *Boletus* spp. were identified as the most preferred for mushroom consumption, followed by *M. procera*, *Leccinum* spp., *C. cibarius*, and *Pleurotus* spp. Consumers from the “regular consumers” segment mainly prefer *Boletus* spp., followed by *Pleurotus* spp., *Leccinum* spp., and *M. procera*. In the “heavy consumers” segment, *Boletus* spp., *C. cibarius*, *Leccinum* spp., *Pleurotus* spp., *Xerocomus* spp., and *M. procera* were the most preferred by consumers. Based on the above results, we can conclude that the most species of edible mushrooms preferred by consumers were identified in the “heavy consumers” segment. The results showed a higher preference for edible mushrooms in the “heavy consumers” segment compared to the other two segments, which may be due to a higher amount of edible mushroom consumption. By applying the Friedman test and the subsequent Nemenyi post-hoc test, we examined the evaluation of the preference of mushroom species in individual segments. Based on the results, it can be concluded that statistically significant differences were identified in the preferences of individual species of mushrooms within all segments. Demsar plots indicated the results of the Nemenyi test, and it was used to graphically represent the confirmation of differences in consumer preferences for eating edible mushrooms for each segment (Figure 1, Figure 2 and Figure 3).

The examination of consumer preferences in the consumption of edible mushrooms was also aimed at finding out where consumers from individual segments most often consume edible mushrooms. In general, it can be stated that consumers from all segments most often consume mushroom meals at home. However, the results of the applied Chi-square test showed statistically significant differences in identifying the most preferred place of edible mushrooms consumption (*p* < 0.0001). It can be concluded that 77.7% of consumers from the “occasional consumers” segment consume mushrooms at home, but more than 10% consume them at family members’ homes and 7.3% in restaurants. For comparison, more than 90% of consumers from the segment “regular consumers” and the segment “heavy consumers” consume mushrooms at home, and less than 5% prefer to consume them with family members. A more detailed specification of the preferred place for edible mushroom consumption, divided into individual segments, is shown in the following Table 8.

The aim of the consumer study was also to identify what options consumers use to obtain edible mushrooms for consumption. Consumers from all segments chose which of the following options for obtaining mushrooms they use, namely, cultivation of edible mushrooms, collection of wild edible mushrooms, buying edible mushrooms, and getting edible mushrooms as a gift. Based on the applied Chi-square test, statistically significant differences between individual segments were identified in all options of obtaining: cultivation of edible mushrooms (*p* < 0.0001), collection of wild edible mushrooms (*p* < 0.0001), buying edible mushrooms (*p* = 0.0019), and getting edible mushrooms as a gift (*p* = 0.0004).

Regarding the issue of mushroom cultivation, it can be stated that consumers from the “heavy consumers” segment (23.4%) have the highest tendency to cultivate mushrooms. Collection of mushrooms is more widespread in the segment “heavy consumers” (83.2%) and “regular consumers” (77.9%). Mushrooms for consumption are also obtained by purchasing them, and mushrooms are most purchased by consumers from the “heavy consumers” segment (62.1%), followed by consumers from the “regular consumers” segment (57.5%), and they are least purchased by consumers from the “occasional consumers” segment (50.2%). The last monitored option of obtaining mushrooms for consumption was getting edible mushrooms as a gift. The results show that mainly consumers from the “occasional consumers” segment get mushrooms as gifts (40.7%). The option of obtaining mushrooms for consumption and their use by consumers from individual segments are specified in more detail in Table 9.

Based on the above results, it can be concluded that consumers from all segments most often obtain mushrooms for consumption by collection or buying. The results showed that 65.2% of consumers from the “occasional consumers” segment are also mushroom pickers. In the behavior of consumers when collecting mushrooms, it was identified that the majority of consumers from the mentioned segment collect mushrooms 2–10 times a year, preferring mixed forests and forest edges for collection. Furthermore, it was found that they prefer to collect 2–5 species, and the most collected are *B. reticulatus*, *B. edulis*, *M. procera*, *Leccinum* spp., and *C. cibarius*. In total, 66% of mushroom pickers from the segment “occasional consumers” prefer to collect mushrooms near their residence at a distance of up to 20 km, and they most often collect up to 1 kg of edible mushrooms per collection. In the “regular consumers” segment, it was identified that almost 80% of mushroom consumers are also wild edible mushroom pickers. The behavior of pickers from the “regular consumers” segment was similar to pickers from the “occasional consumers” segment, but slight differences were identified. More than 55% of pickers from the “regular consumers” segment collect mushrooms 2–10 times a year, while more than a third of pickers collect mushrooms more than 10 times a year. Furthermore, we identified that they collect 2–5 species or 6–10 species of edible mushrooms, and the most preferred are *B. reticulatus*, *B. edulis*, *M. procera*, *Leccinum* spp., *Xerocomus* spp., and *C. cibarius*. Most collectors from the mentioned segment prefer to collect in mixed forests and the edges of forests, which are at a maximum distance of 50 km from their place of residence, and in one collection, they usually collect 1–2 kg of edible mushrooms. Consumers from the “heavy consumers” segment represent intensive mushroom pickers, as 85% of them collect mushrooms. Compared to the other two segments, pickers from the segment “heavy consumers” collect mushrooms 2–10 times per year (37.8%), 11–20 times per year (18.2%), or more than 20 times per year (41.5%). During the year, they usually collect more than 10 species of edible mushrooms and the most preferred are *B. reticulatus*, *B. edulis*, *Xerocomus* spp., *C. cibarius*, *Leccinum* spp., and *M. procera*. For mushroom collection, they prefer mixed forests and forest edges at a distance of more than 50 km (30.0%). During one collection, they mostly collect 1–2 kg of wild edible mushrooms (39.7%) or 2–5 kg of wild edible mushrooms (38.8%).

The second most frequented option of obtaining mushrooms for consumption is their purchase. In each of the examined segments, there is a majority of consumers who buy edible mushrooms. The largest number of consumers who are also buyers is in the “heavy consumers” segment, which can be justified by the higher annual consumption of over 5 kg. Significant differences in the purchasing behavior of consumers between individual segments were not identified, and in general, it can be summarized that mushrooms are a food that is purchased by most consumers only occasionally. The highest frequency of purchase, at least once a week, was identified in the “heavy consumers” segment (27.4%). Fresh mushrooms available in supermarkets are the most purchased by consumers from all segments, and consumers are willing to pay up to EUR 10 for 1 kg of fresh edible mushrooms. Consumers also evaluated the factors determining the purchase and subsequent consumption of mushrooms on a 5-point Likert scale. Based on the results, it can be concluded that the quality and the preferred species of mushroom are the key ones. No statistically significant differences were identified in the evaluation of factors between individual segments. However, differences between the individual factors evaluated by all consumers buying edible mushrooms were identified, and hidden relationships between these factors were examined. Based on the results of the applied categorical principal component analysis (CATPCA), we identified four latent factors determining the purchase of edible mushrooms (Table 10). The first latent factor is mushroom sales marketing, which includes factors such as promotion, packaging attributes, sales promotion, sales location, package size, packaging material, or recommendations. The second factor is the attributes of the product, which consist of freshness, appearance, quality, species of mushroom, or its smell. The third latent factor is the authenticity factor involving the country of origin and the location of collection, which are crucial for the perception of the health aspect of the mushroom. The last latent component is the price factor, consisting of the price of mushrooms, as well as price discounts in retail stores.

Based on the achieved results of the consumer study, it is possible to state that the consumption of edible mushrooms is relatively low among Slovak consumers, especially in the “occasional consumers” segment. On the other hand, however, it is desirable to maintain the level of consumption in the “regular consumers” and “heavy consumers” segments. For this reason, we have identified sources of information that can determine Slovak consumers in the consumption of edible mushrooms. Based on the results of the Kruskal–Wallis test, we state a different assessment of the influence of the following information sources: advertising, professional articles, popularization articles, recommendation from experts, and reportage on mushroom consumption between individual segments (*p* < 0.05). On the other hand, no statistically significant differences were identified in the assessment of the influence of the following sources of information on the consumption of mushrooms between individual segments, namely, in-store sales support, promotional stock flyers and magazines, recommendation from doctors, recommendations from friends, promotion by a well-known personality, social networks, and cultural and social events (*p* > 0.05). Based on the results, we can conclude that the most influential source of information for consumers and support for consumption are primarily recommendations from friends and experts, as well as published professional and popularizing articles related to the consumption of mushrooms. By applying the Friedman test and the subsequent Nemenyi test, we examined the differences in the evaluation of the influence of individual sources of information on the consumption of mushrooms in individual segments. Based on the results, it can be concluded that statistically significant differences were identified in the assessment of the influence of sources of information on the consumption of mushrooms (*p* < 0.0001). Statistically significant differences in the assessment of the impact of individual sources of information on the consumption of mushrooms for each segment are shown graphically in the following graphs (Figure 4, Figure 5 and Figure 6).

## 4. Discussion and Conclusions

Edible mushrooms are classified as healthy and functional foods due to their nutritional benefits, and their consumption is recommended. The results of a study conducted on a sample of 1166 Slovak respondents showed that 88.5% of them are consumers of edible mushrooms, but occasional consumption is predominant. Other studies conducted in different countries have reached similar results, which emphasize the consumption of mushrooms by more than 80% of consumers, but it is important to point out that the consumption of mushrooms is rather occasional [8,22,30]. Ballesterol et al. [39] found that 76.0% of consumers consume mushrooms only once a month or seldomly in a year.

The study further analyzed the consumer behavior of Slovak mushroom consumers, who were divided into three segments depending on the amount of consumption. The first segment represented consumers with a consumption of up to 1.0 kg and was named “occasional consumers”, the second segment grouped consumers with an annual consumption between 1.1 and 5.0 kg and was named “regular consumers”, and the last segment included consumers with consumption higher than 5.0 kg and was named “heavy consumers”. Until now, studies focused on the segmentation of mushroom consumers have already been carried out. Segments were created based on opinions on the selection of available mushrooms and consumer awareness, and three segments were identified, namely, health-conscious consumers, indifferent consumers, and average consumers [9]. Veljović and Krstić [4] divided consumers into segments based on purchasing behavior and willingness to pay for mushrooms and defined three segments, namely, the segment with a high probability of purchasing, the segment with a low probability of purchasing, and the segment with no probability of purchasing.

The results of the consumer study carried out by us further point to the behavior and preferences of Slovak consumers from individual segments when consuming edible mushrooms. It was identified that consumers from the “heavy consumers” segment consume mushrooms most often compared to other segments. By applying statistical methods, differences between individual segments were found in the preference for mushroom consumption as part of meals, evaluation of important reasons for consumption, determination of preferred species of mushrooms for consumption, and determination of preferred place of consumption, as well as the selection of options for obtaining mushrooms for consumption. In general, however, it can be stated that consumers prefer to eat mushrooms in the main meal or soup. Shirur and Shivalingegowda [23] pointed out a preference for eating mushrooms as part of soup and curry in their study. The key reason for the consumption of mushrooms by Slovak consumers is the taste, which is also confirmed by the results of other studies [39]. Slovak consumers’ preferred species of mushrooms for consumption are *Boletus* spp., *Leccinum* spp., and *C. cibarius*. Other studies have shown a high preference for button and oyster mushrooms [14,23]. Slovak consumers most often prepare dishes from mushrooms at home, where they also consume them. From the point of view of the possibilities of obtaining wild mushrooms for consumption, Slovak consumers most often use the collection of mushrooms in mixed forests near their place of residence. Gizaw et al. [40] identified that edible mushrooms are collected from cultivated farm land and in different areas, such as termite nests, uncultivated farm land, and forests in the vicinity, while little collection is done in wood lands. Slovak consumers consider the purchase of mushrooms to be the second most preferred option for obtaining mushrooms for consumption, and they prefer supermarkets and the selection of fresh mushrooms for purchase. Other studies have also identified that the most preferred major place for buying mushroom are retail shops and supermarkets [7,23,41], and consumers look for fresh mushrooms rather than frozen or otherwise processed ones [33]. The purchasing process of Slovak consumers is determined by a number of factors, and the key ones are the quality and species of mushroom. Our results are also confirmed by a study conducted by Gürgen et al. [42], who identified growing type and quality as the most important criteria in the purchasing process. An interesting finding was that no statistically significant differences between the individual segments were identified in the purchasing behavior of Slovak consumers, which can be justified precisely by the fact that it is important for consumers to consume high-quality and healthy mushrooms, regardless of the fact of how many mushrooms they consume annually. In the context of the examined factors, four latent factors determining the purchase and subsequent consumption of mushrooms were identified, namely mushroom sales marketing, product attributes, authenticity factor, and price factor. Mohd Tarmizi et al. [33] applied factor analysis in his study and identified four components determining the purchase of mushrooms and mushroom products, namely, product attributes, health benefits, consumers’ perception, and certification. Regarding the possibilities of obtaining mushrooms for consumption, Christy et al. [29] identified that 85.3% of the respondents like to grow mushrooms for their own consumption.

In general, it can be concluded that despite the annual growing production of cultivated mushrooms, the consumption of mushrooms not only in Slovakia, but also in the world, is relatively low. For this reason, the influence of information sources was examined in order to support the consumption of mushrooms. The results showed that recommendations and articles focused on the necessity of mushroom consumption, as well as the health benefits of mushrooms, have the most significant impact on increasing edible mushroom consumption in Slovakia. A study conducted by Mohd Tarmizi et al. [33] showed that for 69.5% of consumers, the easiest way to obtain information about products related to mushrooms is electronic media, such as television, radio, the Internet, or social media, and also printed media, such as magazines, newspapers, and brochures. Research carried out by Tejera et al. [43] additionally found that more than half of the respondents were unaware of the health benefits of mushroom consumption.

The results of the consumer study are beneficial for businesses focused on business with mushrooms and are applicable in the creation of marketing strategies. Furthermore, the results of the study include a new perspective on the behavior and preferences of consumers when consuming mushrooms, divided into segments in terms of the amount of mushrooms consumed, thus filling a gap in the scientific research area. Based on the identified results, we propose to increase consumer awareness of the need to consume edible mushrooms through various marketing communication methods and by educating consumers through popularization and expert articles. The results of the study can also be applied in the creation of programs promoting and supporting healthy eating habits in society.

It is also important to point out the limitations of the research paper. The key limitation is the application of self-reported measures related to consumption patterns and the subjective evaluation of the amount consumed, as well as the complex behavior and preferences in the consumption of edible mushrooms. The second limitation is the territoriality of the research; the questionnaire survey was carried out at the national level of the Slovak Republic. Future research directions could be focused on international comparative studies that explore the similar or different behavior of consumers when consuming edible mushrooms.

## Figures and Tables

**Figure 1 foods-12-00657-f001:**
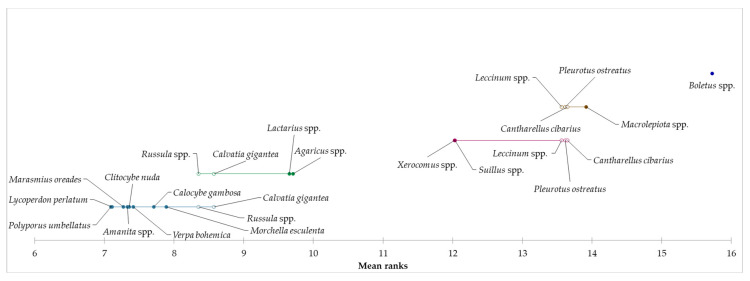
Differences in the evaluation of preferred species of mushrooms in the “occasional consumers” segment.

**Figure 2 foods-12-00657-f002:**
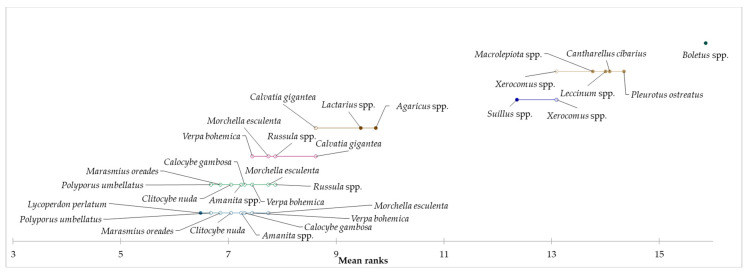
Differences in the evaluation of preferred species of mushrooms in the “regular consumers” segment.

**Figure 3 foods-12-00657-f003:**
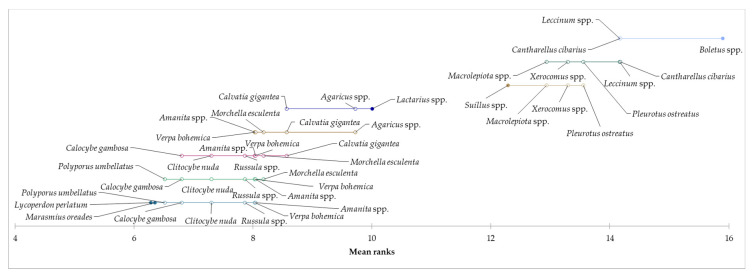
Differences in the evaluation of preferred species of mushrooms in the “heavy consumers” segment.

**Figure 4 foods-12-00657-f004:**
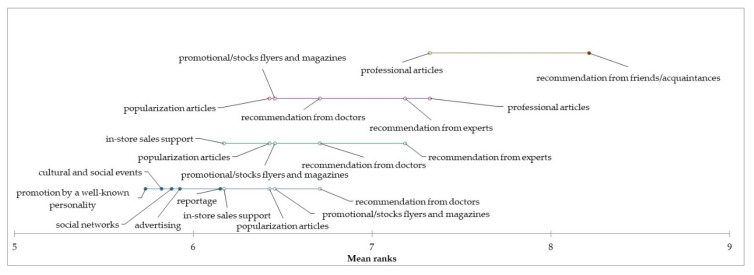
Differences in the evaluation of the influence of information sources on the edible mushrooms consumption in the “occasional consumers” segment.

**Figure 5 foods-12-00657-f005:**
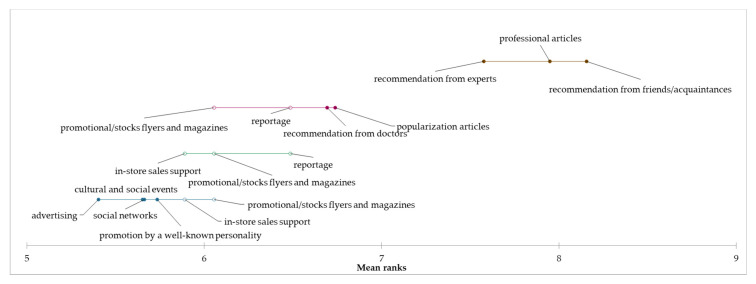
Differences in the evaluation of the influence of information sources on the edible mushrooms consumption in the “regular consumers” segment.

**Figure 6 foods-12-00657-f006:**
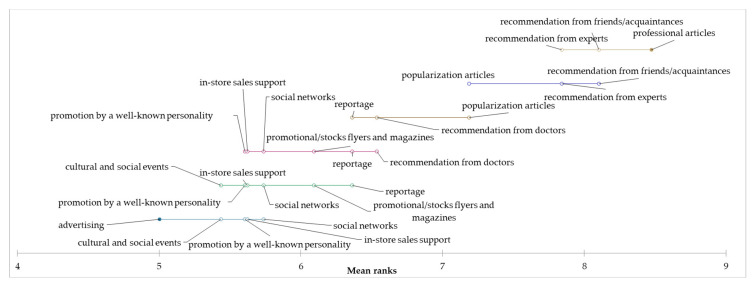
Differences in the evaluation of the influence of information sources on the edible mushrooms consumption in the “heavy consumers” segment.

**Table 1 foods-12-00657-t001:** Socio-demographic characteristics of respondents.

Socio-Demographic Characteristics		Whole Sample		Sample of Mushrooms Consumers	
		*n*	%	*n*	%
Gender	men	508	43.6	466	45.2
	women	658	56.4	566	54.8
Age category	<25 years	378	32.4	304	29.5
	25–39 years	330	28.3	287	27.8
	40–54 years	305	26.2	290	28.1
	>54 years	153	13.1	151	14.6
Residence	urban	628	53.9	566	54.8
	rural	538	46.1	466	45.2
Members in households	1–2 members	360	30.9	325	31.5
	3–4 members	618	53.0	544	52.7
	5 and more members	188	16.1	163	15.8
Education	elementary	7	0.60	5	0.43
	secondary	457	39.2	397	38.5
	university	702	60.2	630	61.0
Economic status	employed	572	49.1	526	51.0
	entrepreneur	71	6.09	67	6.47
	student	421	36.1	338	32.8
	pensioner	58	4.97	58	5.61
	maternity leave	18	1.54	17	1.62
	unemployed	26	2.23	26	2.50

**Table 2 foods-12-00657-t002:** Socio-demographic characteristics of individual segments of edible mushrooms consumers.

Socio-Demographic Characteristics		Occasional Consumers *n* = 273		Regular Consumers *n* = 503		Heavy Consumers *n* = 256	
		*n*	%	*n*	%	*n*	%
Gender	men	99	36.3	235	46.7	132	51.6
	women	174	63.7	268	53.3	124	48.4
Age category	<25 years	102	37.4	141	28.0	61	23.8
	25–39 years	76	27.8	147	29.2	64	25.0
	40–54 years	70	25.6	142	28.2	78	30.5
	>54 years	25	9.2	73	14.5	53	20.7
Residence	urban	128	46.9	278	55.3	160	62.5
	rural	145	53.1	225	44.7	96	37.5
Members in households	1–2 members	65	23.8	167	33.2	93	36.3
	3–4 members	156	57.1	263	52.3	126	49.2
	5 and more members	53	19.4	73	14.5	37	14.5
Education	elementary	1	0.4	0	0.0	4	1.6
	secondary	117	42.9	185	36.8	95	37.1
	university	155	56.8	318	63.2	157	61.3
Economic status	employed	131	48.0	257	51.1	138	53.9
	entrepreneur	13	4.8	33	6.6	21	8.2
	student	111	40.7	159	31.6	68	26.6
	pensioner	8	2.9	29	5.8	21	8.2
	maternity/parental leave	6	2.2	8	1.6	3	1.2
	unemployed	4	1.5	17	3.4	5	2.0

**Table 3 foods-12-00657-t003:** Frequency of edible mushrooms consumption in individual segments of edible mushrooms consumers.

	Occasional Consumers		Regular Consumers		Heavy Consumers	
	*n*	%	*n*	%	*n*	%
several times a week	0	0.0	5	1.0	24	9.4
1 time a week	3	1.1	67	13.3	79	30.9
1–2 times a month	64	23.4	245	48.7	123	48.0
occasionally	206	75.5	186	37.0	30	11.7

**Table 4 foods-12-00657-t004:** Differences in the evaluation of preferred mushrooms meals in the “occasional consumers” segment.

Sample	Frequency	Sum of Ranks	Mean of Ranks	Groups		
salty snacks	273	751.500	2.753	A		
appetizer	273	760.000	2.784	A		
salad	273	778.500	2.852	A		
side dish	273	1112.500	4.075		B	
flavoring of meals	273	1172.500	4.295		B	
main meal	273	1526.000	5.590			C
soup	273	1543.000	5.652			C

**Table 5 foods-12-00657-t005:** Differences in the evaluation of preferred mushrooms meals in the “regular consumers” segment.

Sample	Frequency	Sum of Ranks	Mean of Ranks	Groups		
salty snacks	503	1294.000	2.573	A		
salad	503	1360.500	2.705	A		
appetizer	503	1386.000	2.755	A		
side dish	503	2146.000	4.266		B	
flavoring of meals	503	2150.000	4.274		B	
soup	503	2844.500	5.655			C
main meal	503	2903.000	5.771			C

**Table 6 foods-12-00657-t006:** Differences in the evaluation of preferred mushrooms meals in the “heavy consumers” segment.

Sample	Frequency	Sum of Ranks	Mean of Ranks	Groups		
salad	256	665.000	2.598	A		
salty snacks	256	688.000	2.688	A		
appetizer	256	713.000	2.785	A		
side dish	256	1075.500	4.201		B	
flavoring of meals	256	1088.500	4.252		B	
soup	256	1455.000	5.684			C
main meal	256	1483.000	5.793			C

**Table 7 foods-12-00657-t007:** Reasons for edible mushrooms consumption in individual segments of edible mushrooms consumers.

	Occasional Consumer		Regular Consumer		Heavy Consumer	
	*n*	%	*n*	%	*n*	%
taste	188	68.9	325	64.6	175	68.4
habit/custom	54	19.8	84	16.7	39	15.2
recommendation	5	1.8	4	0.8	0	0.0
part of a rational diet	25	9.2	73	14.5	22	8.6
life style	1	0.4	17	3.4	20	7.8

**Table 8 foods-12-00657-t008:** Places of edible mushrooms consumption in individual segments of edible mushrooms consumers.

	Occasional Consumers		Regular Consumers		Heavy Consumers	
	*n*	%	*n*	%	*n*	%
at home	212	77.7	462	91.8	241	94.1
canteens at school/workplace	11	4.0	8	1.6	2	0.8
restaurant	20	7.3	13	2.6	3	1.2
at family members	30	11.0	20	4.0	10	3.9

**Table 9 foods-12-00657-t009:** Options for obtaining mushrooms for consumption in individual segments of edible mushrooms consumers.

		Occasional Consumers		Regular Consumers		heavy Consumers		*p*-Value
		*n*	%	*n*	%	*n*	%	
cultivation of edible mushrooms	yes	12	4.4	56	11.1	60	23.4	<0.0001
	no	261	95.6	447	88.9	196	76.6	
collection of wild edible mushrooms	yes	178	65.2	390	77.5	214	83.6	<0.0001
	no	95	34.8	113	22.5	42	16.4	
buying edible mushrooms	yes	143	52.4	316	62.8	168	65.6	0.003
	no	130	47.6	187	37.2	88	34.4	
getting edible mushrooms as a gift	yes	111	40.7	148	29.4	77	30.1	0.004
	no	162	59.3	355	70.6	179	69.9	

**Table 10 foods-12-00657-t010:** Factor loadings from Categorical Principal Component Analysis (CATPCA)—Factors determining edible mushrooms purchase and consumption.

Factors	Dimensions			
	1	2	3	4
promotion	0.776	0.139	0.241	0.172
packaging	0.733	0.238	0.116	0.033
sales support	0.729	0.164	0.314	0.115
point of sale	0.694	0.045	0.355	0.058
package size	0.651	0.325	0.117	0.187
packaging material	0.627	0.240	0.247	0.115
recommendations	0.565	0.051	0.443	0.158
freshness	0.133	0.834	0.152	0.072
appearance	0.088	0.775	0.121	0.099
quality	0.035	0.746	0.219	0.050
species of mushrooms	0.079	0.654	0.061	0.044
smell	0.152	0.620	0.435	0.103
location of collection	0.307	0.131	0.812	0.025
country	0.217	0.289	0.742	0.077
health	0.219	0.185	0.738	0.129
price	0.058	0.109	0.042	0.877
price discount	0.390	0.086	0.033	0.753

## Data Availability

The data are available from the corresponding author.

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
