# Peer review of "Exploring Consumer Behavior and Preferences towards Edible Mushrooms in Slovakia"

_foods, 2023, doi:10.3390/foods12030657_

Round 1

Reviewer 1 Report

Dear Authors,

I read your paper with a great pleasure. I highly appreciate your ability to assess the issue at various levels, as well as a multidirectional view. The work has a clear layout and the right length. The following parts are logical and do not raise doubts. A specific comments are present below:

1.     A huge problem during reading of this paper is the lack of citations in square brackets, which requires correction throughout the work (for example please look on page 2, lines 63 and 67). A form of references presentation with names of the authors but without the year when paper was published is also a specific problem.

2.     Please remove ,,%” near particular values in Tables 1 and 2 – there is an information about the unit. It may be worth standardizing Table 6, 8 and 9 with Tables 1 and 2 (,,n” and ,,%”)? There are two Tables 8 (please correct the second into Table 9). Table 9 will be a new Table 10.

3.     I suggest to improve of the language, e.g. page 15, line 517: ,,between 1.1 and 5.0 kg” instead ,,between 1.1 kg and 5.0 kg”.

4.     ,,In general, it can be concluded that the consumption of mushrooms not only in Slovakia, but also in the world is relatively low” – It's very important what you write about. Nevertheless, there is a steady increase in mushroom production in the world, hence I suggest putting this valuable statement in concrete realities. It is great that you emphasize this important fact, because mushrooms are extremely valuable and still marginalized by many people.

The comments presented are only guidelines that will improve your valuable work and do not detract from my high substantive assessment. I suggest a minor revision.

Reviewer 2 Report

The paper entitled “Exploring consumer behavior and preferences towards edible mushrooms in Slovakia” is very interesting, especially because supporting consumption of the mushrooms, as a food with high nutritional properties.

Introduction

The introduction is too long; it should be shorter and more concise.

All references should be separated by commas.

Materials and methods

Lines 204 – 243 I suggest to present this part in table, it will be more precise and clear

Results

I suggest explaining which results are in which table before explain the results (lines 285-287 is better to be on the begging of the paragraph)

Discussion and Conclusion

Line 511 I do not understand what 82230 means?

Lines 513-518 were already explain in Results

Discussion and Conclusion part can be more concise.

Reviewer 3 Report

Overall, the topic proposed by the manuscript ” Exploring consumer behavior and preferences towards edible mushrooms in Slovakia” is of interest to the readers and in line with the journal’s aims and scope. The title can be suitable, to be more attractive while the Abstract, Introduction, Materials and Methods, Results and Discussion and Conclusions are well structured, clear and concise. I think the literature is just enough.

The main purpose of this paper is to identify the behavior and preferences of consumers in the consumption of edible mushrooms in Slovakia. The introduction and references are sufficient and especially relevant for deepening the problem addressed.

The methodology seems to be good enough. The sample of 1,166 respondents is representative for the studied area (Slovak Republic). Two research questions were formulated (RQ1, RQ2). Using statistical methods like Kruskal-Wallis H test, Chi-square test of independence, Friedman test, as well as categorical principal component analysis, differences in consumer behavior were examined in three identified segments created based on the amount of consumption of edible mushrooms.

The results are clear and understandable to the readers.

Tables 1 - 7 are intuitive.

Figures 1-6 are welcome. Pay attention to the page layout according to the rigors of the journal.

Attention to the doubling of the numbering of Tables 8 (L387) and 9 (L410).

I consider that the results underlined in this material are not innovative, but they are sufficiently eloquent.The conclusions are as relevant as possible and flow logically and intuitively from the analysis of the paper.

The limitations presented at the end of the conclusions are welcome.

The references respect the journal's specific editorial rules.
